# Virus detection via programmable Type III-A CRISPR-Cas systems

Sagar Sridhara[1,4,5], Hemant N. Goswami [1,5], Charlisa Whyms [2], Jonathan H. Dennis[3] & Hong Li [1,2✉]

Among the currently available virus detection assays, those based on the programmable CRISPR-Cas enzymes have the advantage of rapid reporting and high sensitivity without the requirement of thermocyclers. Type III-A CRISPR-Cas system is a multi-component and multipronged immune effector, activated by viral RNA that previously has not been repurposed for disease detection owing in part to the complex enzyme reconstitution process and functionality. Here, we describe the construction and application of a virus detection method, based on an in vivo-reconstituted Type III-A CRISPR-Cas system. This system harnesses both RNA- and transcription-activated dual nucleic acid cleavage activities as well as internal signal amplification that allow virus detection with high sensitivity and at multiple settings. We demonstrate the use of the Type III-A system-based method in detection of SARS-CoV-2 that reached 2000 copies/µl sensitivity in amplification-free and 60 copies/µl sensitivity via isothermal amplification within 30 min and diagnosed SARS-CoV-2-infected patients in both settings. The high sensitivity, flexible reaction conditions, and the small molecular-driven amplification make the Type III-A system a potentially unique nucleic acid detection method with broad applications.

[1] Institute of Molecular Biophysics, Florida State University, Tallahassee, FL 32306, USA. [2] Department of Chemistry and Biochemistry, Florida State University, Tallahassee, FL 32306, USA. [3] Department of Biological Science, Florida State University, Tallahassee, FL 32306, USA. [4] Present address: Department of Medical Biochemistry and Cell Biology, University of Gothenburg, Gothenburg 40530, Sweden. [5] These authors contributed equally: Sagar Sridhara, Hemant N. Goswami. ✉email: hong.li@fsu.edu

Human infectious diseases constitute a broad class of diseases caused by microorganisms such as bacteria and viruses transmitted via food, air, body fluids, and physical contact[1]. Some of these diseases are highly contagious and broadly affect human health. Historically, several outbreaks namely Spanish flu and Swine flu caused by H1N1 influenza virus[2–4], Acquired Immunodeficiency Syndrome (AIDS) caused by Human Immunodeficiency virus (HIV)[5], Zika disease caused by the Zika virus[6], Ebola Virus Disease (EVD) caused by the Ebola virus[7], Severe Acute Respiratory Syndrome (SARS) caused by SARS coronavirus (SARS-CoV) and Middle East Respiratory Syndrome (MERS) caused by MERS coronavirus (MERS-CoV) have wreaked havoc on general population and had widespread economic implications in the world[8,9]. The global outbreak of Coronavirus disease 2019 (COVID-19) caused by a novel coronavirus named severe acute respiratory syndrome coronavirus 2 (SARS-CoV-2) is the most recent and ongoing public health emergency, transmitting from human to human at very high pace across countries[10–12]. The recent emergence of genetic variants of COVID-19 across the world has further increased the risk of mortality[13–17]. One of the confounding challenges posed by this pandemic and previous infectious diseases has been the availability of rapid testing of the virus, which could limit the spread of the disease.

Respiratory virus infections are generally diagnosed by the presence of either virus-derived antigens, patient immune responses, or nucleic acid materials[18,19]. Antigen-based tests, although effective and rapid, requires prior manufacture of antibodies to be incorporated into the test and usually lack high sensitivity. Serology and other adaptive immune response tests are mature technology but do not distinguish past from active infections. The traditional nucleic acid-based detection, based on Polymerase Chain Reaction (PCR) makes use of nucleic acid amplification methods that has many advantages over antigen-based and serological tests. It is quantitative, highly sensitive and can be made available prior to the onset of a potential pandemic. The recently developed Clustered, Regularly Interspaced, Short Palindromic Repeat (CRISPR) and CRISPR-Associated (Cas) detection methods[20] further revolutionized nucleic acid test. The CRISPR-Cas-based detection has shown high sensitivity and accuracy without requiring the thermocycler used in PCR. In addition, the CRISPR-Cas-based detection is inexpensive, mobile, and rapid, making it deployable as point-of-care testing to locations without sophisticated biochemistry equipment. Current CRISPR-Cas based diagnostic technologies employ mainly Cas12 and Cas13 enzymes that possess either viral DNA or RNA-stimulated collateral DNase or RNase activities, respectively[20]. The Cas13-based methods[21–23] detect the viral RNA by RNA-stimulated cleavage of RNA probes. The Cas12-based methods[22,24,25], on the other hand, detect the viral DNA by DNA-stimulated cleavage of DNA probes. Since the binding constants of Cas13 and Cas12 for their respective targets are typically higher than the potential viral titers in patient samples[24,26], direct detection by the CRISPR enzymes would not be possible without additional assistance. The use of isothermal amplification[24,27], rapid PCR amplification[28], multiplex targeting[29], tandem nucleases cleavage[22,30], and autocatalytic looping[31] are some of the example strategies employed in CRISPR-based detection.

The Type III-A CRISPR-Cas system, or Csm, is a viral RNA or transcription-activated ribonucleoprotein enzyme system that comprises four enzymatic activities: (1) specific cleavage of the viral transcript by the Csm3 subunit, (2) collateral DNase by the HD domain of the Csm1 subunit, (3) cyclic oligoadenylate (cOA) synthesis by the Csm1 GGDD motif, and (4) cOA-activated collateral RNase by the ancillary enzyme Csm6[32–34] (Fig. 1a). The dual collateral activities from the DNase and the cOA-amplified RNase of the Type III-A system could offer a unique system for versatile virus detection. Furthermore, the inherent responses of Csm to either viral RNA or its transcription makes the Csm-based detection suitable for both RNA and DNA viruses. Importantly, the viral activated RNase activity can be disproportionally amplified through supplied oligonucleotides. Despite these promises, it is more challenging to realize such multicomponent systems as an off-the-shelf virus detection tool than their single-subunit counterparts owing to the challenges in enzyme production. Previously, we successfully reconstituted the active *Lactococcus lactis* Csm (LlCsm) complex from an all-in-one expression plasmid in two simple chromatography steps with high yield[35]. Here, we demonstrate the concept of a Csm-derived virus detection system that we termed MORIARTY for Multipronged, One-pot, target RNA-Induced, Augmentable, Rapid, Test sYstem. We further showed the effectiveness of MORIARTY in rapid detection of SARS-CoV-2 virus with high sensitivity, thereby offering a Class I CRISPR-Cas based viral diagnostic.

## Results

**MORIARTY simultaneously and cumulatively detects DNase and RNase fluorescence.** The LlCsm effector complex was conveniently produced using an all-in-one codon-optimized expression plasmid encoding all Csm subunits and the crRNA (Fig. 1a and Supplementary Table 1). The co-purified LlCsm ribonucleoprotein (RNP) cleaves single-stranded DNA (ssDNA) and polymerize adenosine triphosphate (ATP) to generate cyclic oligoadenylates, primarily $cOA_6$, upon binding to its cognate target RNA (CTR) but not noncognate target RNA (NTR) or self RNA (Fig. 1a)[35]. The generated $cOA_6$ second messenger subsequently activates a nonspecific ssRNA cleavage activity in the ancillary protein LlCsm6, expressed and purified separately (Fig. 1a)[35]. To harness the detectable viral RNA-stimulated DNase and RNase activities, we constructed two fluorescence reporters, an RNA oligo flanked by a fluorophore-quencher pair and a DNA oligo flanked by either the same or a different fluorophore-quencher pair (Fig. 1a). We reconstituted the DNase and RNase activities by adding a model target RNA complementary to the guide RNA (Fig. 1a and Supplementary Tables 2–4) to a reaction mixture containing LlCsm, LlCsm6, ATP, and the two fluorescence probes (Fig. 1b, c and Supplementary Fig. 1).

We first tracked the dual functionality of LlCsm system simultaneously by using the DNA reporter flanked by Alexa594N fluorophore-quencher (hereafter referred to as DNA-Alexa) and the RNA reporter flanked by the Fluorescein (FAM) fluorophore-quencher (hereafter referred to as RNA-FAM) with absorption/emission wavelength of 570/630 nm and 480/530 nm respectively (Supplementary Table 4). Such a setup enabled us to detect both activities of LlCsm1 in parallel via real-time intensity rises in two separate fluorescence channels stimulated by a model target RNA (Fig. 1b and Supplementary Fig. 1). As the DNase activity of LlCsm1 strictly requires $Mn^{2+}$[36,37] while cOA synthesis is supported by either $Mg^{2+}$ or $Mn^{2+}$[34,38,39], we monitored fluorescence signals under different metal ion conditions. In the presence of $Mg^{2+}$ and ATP, strong RNA-FAM but no DNA-Alexa fluorescence signal was observed (Fig. 1b and Supplementary Fig. 1). A combination of $Mn^{2+}$ and low ATP concentration, on the other hand, yielded both DNA-Alexa and RNA-FAM signals (Fig. 1b and Supplementary Fig. 1), and under the same condition with DNA-Alexa replaced by DNA-FAM, the total fluorescence was augmented (Fig. 1b). Consistently, there was no RNA-FAM signal without ATP or no DNA-Alexa signal without $Mn^{2+}$ (Supplementary Fig. 1). We further confirmed the source of either fluorescence signal with the HD domain mutant of

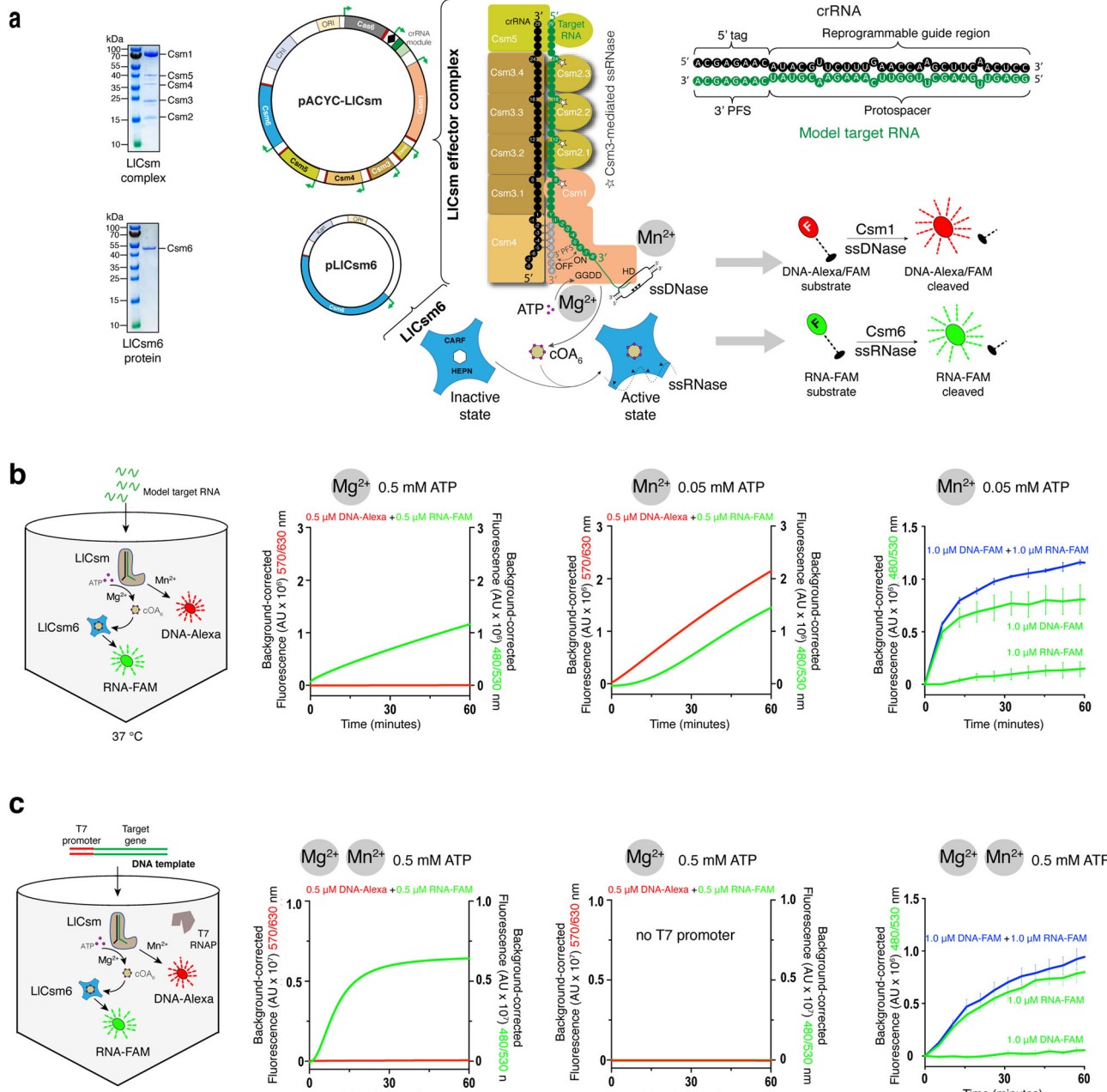

**Fig. 1 Design concept and construction of MORIARTY. a** Expression-plasmids, purified Csm components and schematic representation of the LlCsm/LlCsm6 CRISPR-Cas system and the model target RNA. Target RNA-induced cleavage by Csm3, DNase by Csm1, cOA$_6$ synthesis by Csm1 and cOA$_6$-stimulated RNase by Csm6 are indicated. The target RNA with 3′ protospacer flanking sequence (3′ PFS) that activates LlCsm is shown in green and that inhibits LlCsm is shown in gray. The DNA fluorescent probes (DNA-Alexa, red) used in the study are substrates for LlCsm1-stimulated DNase activity. The RNA fluorescent probe (RNA-FAM, green) is a substrate for cOA$_6$ stimulated LlCsm6-mediated RNase activity. The cleavage of DNA-Alexa is indicated by the rise in fluorescence intensity at 570/630 nm excitation/emission wavelength. The cleavage of DNA-FAM/RNA-FAM is indicated by the rise in fluorescence intensity at 480/530 nm excitation/emission wavelength. **b** Scheme and components for one-pot amplification-free MORIARTY in 33 mM Tris acetate pH 7.6 at 32 °C, 66 mM potassium acetate, 0.5 μM DNA-Alexa (or DNA-FAM), 0.5 μM RNA-FAM, 250 nM LlCsm effector complex, 1.0 nM LlCsm6 and a combination of divalent ion and ATP. All reactions were stimulated by 500 nM target RNA (or water) at 37 °C. Left panel, 10 mM MgCl$_2$ and 0.5 mM ATP; Middle panel, 10 mM MnCl$_2$ and 0.05 mM ATP; Right panel, same as middle panel with DNA-Alexa replaced by DNA-FAM either alone (green) or in combination with RNA-FAM (blue). RNA-FAM alone (also green) was included to access signal additiveness. Error bars are indicated for experiments performed in two independent replicates. All signals were background-corrected using the signal with water as the stimulator.
**c** Scheme and components for one-pot T7 MORIARTY in 30 mM K-HEPES pH 7.6, 2 mM Spermidine, 0.01% Triton X-100, 17 mM MgCl$_2$, 0.5 μM DNA-Alexa (or DNA-FAM), 0.5 μM RNA-FAM, 250 nM LlCsm effector complex, 1.0 nM LlCsm6, 0.5 mM rNTPs, 10 mM TCEP, and 60 μg/mL T7 RNA polymerase. All reactions were stimulated by 500 nM target DNA (or water) at 37 °C. Left panel, additional 0.5 mM Mn$^{2+}$ was used; Middle panel, DNA template without T7 promoter sequence was used as the stimulator; Right panel, same as left panel with DNA-Alexa replaced by DNA-FAM either alone (green) or in combination with RNA-FAM (blue). RNA-FAM alone (also green) was included to access signal additiveness. Error bars are indicated for experiments performed in two independent replicates. All signals were background-corrected using the signal with water as the stimulator.

LlCsm1 (H13A/D14N) and the HEPN domain mutant of LlCsm6 (LlCsm6 R355A) that removed the DNase and RNase, respectively (Supplementary Fig. 1). These results show that MORIARTY gives rise to two target-induced signals detectable under multiple buffer conditions.

The RNA-guided Type III-A CRISPR-Cas systems are known to be directly activated by viral RNA transcription[40,41]. We wondered if a DNA template with the T7 promoter sequence that encodes the viral RNA may be used as the stimulator to the LlCsm system in presence of T7 transcription components in addition to MORIARTY reactants, henceforth referred to as T7-MORIARTY. Under the reaction condition containing $Mg^{2+}$ and ATP, RNA-FAM but not DNA-Alexa yielded strong fluorescence due to the absence of $Mn^{2+}$ (Supplementary Fig. 1). No fluorescence was observed with the DNA stimulator lacking the T7 promotor sequence, indicating a strict dependence of T7-MORIARTY on transcription (Fig. 1c). As $Mg^{2+}$ is essential to while high $Mn^{2+}$ concentrations inhibit T7 transcription[42], we wondered if a small amount of $Mn^{2+}$ can harness both the RNase and the DNase signals. We supplemented the T7-MORIARTY reaction with 0.5 mM $MnCl_2$ and observed a small but definitive additiveness from both DNA-FAM and RNA-FAM (Fig. 1c). T7-MORIARTY provides the option of detecting nucleic acids at low concentrations coupled with pre-amplification.

**Optimization of MORIARTY for amplification-free detection of SARS-CoV-2 viral RNA**. To make MORIARTY effective without amplification, we optimized it for detection of SARS-CoV-2 responsible for the COVID-19 pandemic. We reprogrammed the 29mer CRISPR RNA protospacer (Fig. 2a and Supplementary Tables 2 and 3) to detect the Spike (S) gene of SARS-CoV-2 (nucleotides 22280–22308, NCBI MT801051.1) (Supplementary Tables 2 and 3) and purified the LlCsm RNP (hereafter LlCsm_S0). The choice of the targeting region within the S gene was made such as to ensure that the 3′-protospacer flanking sequence (3′-PFS) of the viral target RNA would remain non-complementary to the 5′-tag of the LlCsm crRNA[34,37,39] to mimic that for CTR (Fig. 2a). To evaluate if LlCsm_S0 would successfully detect viral RNA without amplification, we first optimized MORIARTY with a synthetic target RNA (S0_CTR) or an in vitro transcribed S mRNA (hereafter S_IVT_RNA) (Supplementary Tables 2 and 3) at femto- to pico-Molar (fM-pM) concentrations with the objective to distinguish the target signal from the water signal. Specifically, we varied the concentrations of LlCsm6 and ATP to take advantage of the powerful signal amplification through cOA6 (Supplementary Fig. 2). We also varied concentrations of the fluorescence probes (0.5–2 μM) and explored the best combination of metal ions ($Mg^{2+}$, $Mn^{2+}$, and $Mg^{2+}$ plus $Mn^{2+}$) (Supplementary Fig. 2). Lastly, we designed LlCsm_S0_D30A that harbors D30A mutation in its Csm3 subunit to take advantage of the sustained DNase and the cOA synthesis activities (Supplementary Fig. 2)[35].

We found that elevating both LlCsm6 and ATP concentrations is the most effective in increasing the signal contrast. The change from 0.5 mM to 1.5 mM ATP increased the sensitivity by nearly 1000-fold (Supplementary Fig. 2a). The improved signal with elevated LlCsm6 and ATP is a result of their roles in cOA6-mediated signal amplification. Noteworthy, we observed that the LlCsm6 possesses the ring nuclease activity that degrades cOA6 whilst sensing it by a mechanism currently not known (Supplementary Fig. 2a). We envision that if this activity can be inhibited without affecting the RNase activity, LlCsm6 could potentially further improve the sensitivity. With an appropriate combination of other components, we found that $Mg^{2+}$ and $Mn^{2+}$ impact the sensitivity differently. Under the $Mn^{2+}$ condition that both DNase and cOA6 synthesis are active, 5 fM

S_IVT_RNA could be detected with high confidence when comparing to water if ATP was kept at a low concentration (Supplementary Fig. 2b). Similarly, under the $Mg^{2+}$ condition when DNase activity is negligible, 5 fM of S_IVT_RNA was also confidently detected in combination with high ATP concentrations (Supplementary Fig. 2b, c).

To further increase the detection sensitivity, we employed a multiplex targeting strategy by including two additional LlCsm effectors that target S mRNA nucleotides 24702–24730 (LlCsm_S7) and nucleotides 25061–25089 (LlCsm_S8), respectively (Supplementary Tables 2 and 3). We optimized LlCsm_S7 and LlCsm_S8 individually and in combination with LlCsm_S0 and further improved the detection sensitivity by 10-fold in a $Mg^{2+}$ condition by using all three RNPs (Fig. 2a and Supplementary Fig. 2c). When multiplexing was applied to independently quantified PCR control RNA extracted from heat-inactivated SARS-CoV-2 virus (BEI NR-52347) with known viral titers (50,000 cp/μL), we achieved detection limit of 2000 copies/μL (Fig. 2b).

Finally, we applied multiplexing MORIARTY to human patient nasopharyngeal swab samples obtained from Florida State University/Tallahassee Memorial Hospital COVID-19 testing center. We extracted RNA from six patients and split the RNA for simultaneous detection with MORIARTY and an FDA approved q-RT-PCR procedure, respectively. For patient samples identified as positive and with low Ct values by q-RT-PCR, the amplification-free MORIARTY also yielded positive identification with *p*-values < 0.0070 (Fig. 2c). However, patient samples identified as positive with high Ct values were not diagnosed at the current MORIARTY setting, indicating further optimization is needed to achieve detection completely amplification-free (Fig. 2c).

**Optimization of amplification-coupled MORIARTY for attomolar detection of SARS-CoV-2 virus**. We next examined if the sensitivity of MORIARTY can be further improved beyond 2000 copies/μL when it is coupled with a pre-amplification step using reverse transcription and recombinase-polymerase amplification (RT-RPA)[21,43]. In this regard, viral RNA is first reverse transcribed by a reverse transcriptase (RT) followed by double stranded DNA synthesis from a T7 promoter-containing primer and a downstream primer via three recombinase-polymerase amplification (RPA) enzymes: a recombinase, single-stranded DNA-binding protein and strand-displacing polymerase[44] (Fig. 3a). We took advantage of the RT-RPA protocols and primer designs optimized previously for the detection of the S gene of SARS-CoV-2[45] (Supplementary Table 5) and tested the sensitivity of MORIARTY to RT-RPA reaction products.

To reach the best reaction condition used for RT-RPA MORIARTY, we first optimized T7 MORIARTY without including the RT-RPA components. We stimulated LlCsm_S0_D30A/LlCsm6 with T7 promoter-containing DNA that encodes S0_CTR (Supplementary Table 5). While the amplification-free assay supports detection under multiple conditions, the T7-based detection is limited to those favoring transcription. We thus focused on LlCsm6, ATP, probe concentrations in buffers containing $Mg^{2+}$. Similar to amplification-free MORIARTY, increasing LlCsm6 concentration to ~250 nM and probe concentration to 1.5 μM resulted in a significant increase in signal contrast (Supplementary Fig. 3). Under the optimized T7 MORIARTY condition, we addressed target specificity by using DNA substrates bearing single or double mismatches to the guide region of the crRNA (Supplementary Fig. 3a). We found that although single mismatches in the region closer to the 5′-tag provided ~40% reduction, double mismatches caused a near 90% reduction in signal contrast (Supplementary Fig. 3a), making it possible for MORIARTY distinguish mutant virus strains if a synthetic mismatch is programmed into the crRNA.

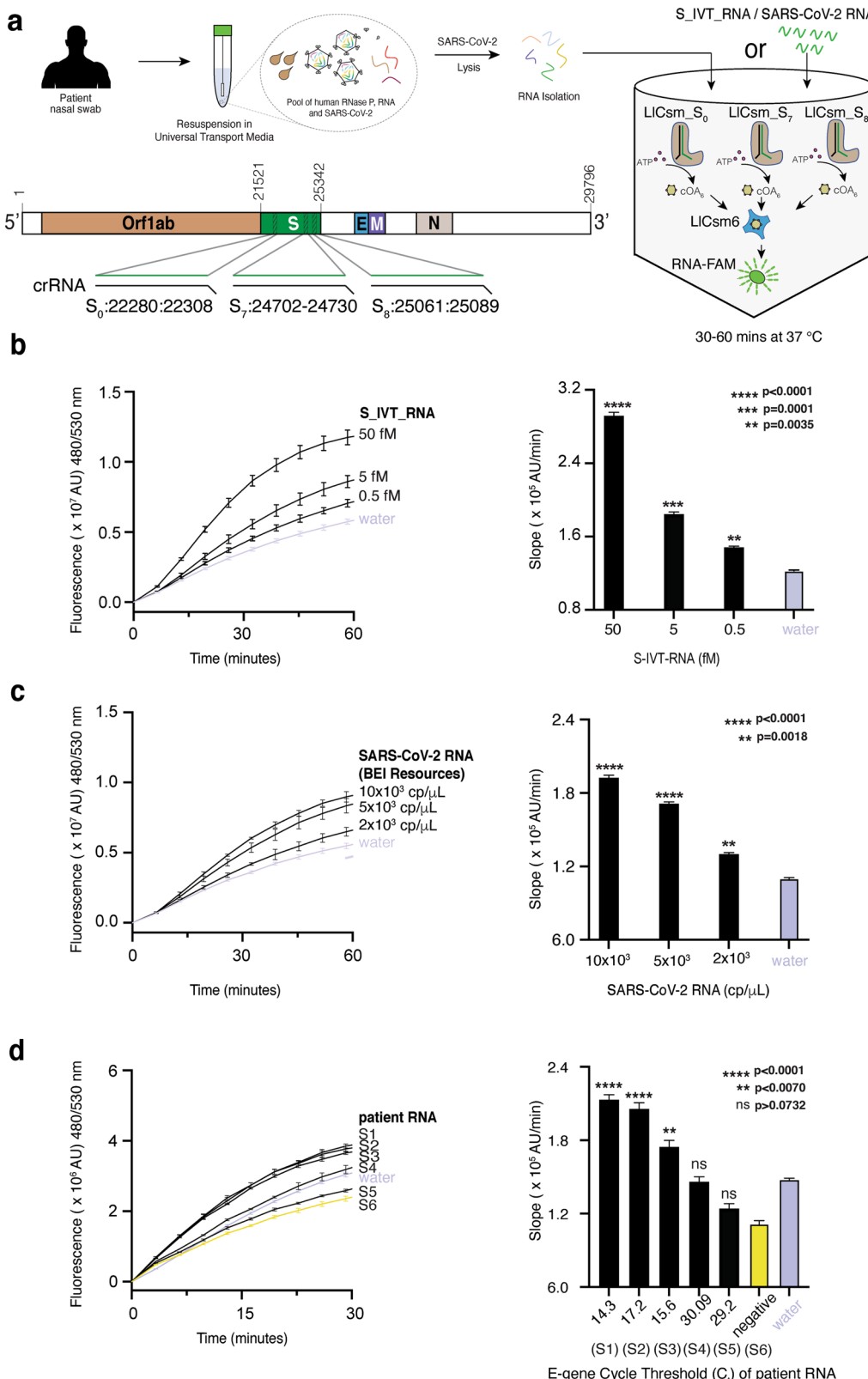

We proceeded to detect S_IVT_RNA by performing RT-RPA and used its product to stimulate LlCsm_S0_D30A/LlCsm6 and further optimized LlCsm6, ATP, divalent ions, and probe concentrations (Supplementary Fig. 3b, c). A 50 µL RT-RPA reaction was first performed using 5 µL S_IVT_RNA at varying concentrations by incubating the reaction mix at a 42 °C water bath for 25 min. 12.5 µL of the RT-RPA reaction product was

then used to stimulate a 25 µL LlCsm_S0_D30A/LlCsm6 reaction. Under the optimized reaction condition, a clear rise in total fluorescence above water was observed with as low as 1 fM S_IVT_RNA (~600 copies/µL) (Supplementary Fig. 3b).

We next applied RT-RPA-coupled MORIARTY to the detection of Quantitative PCR control RNA extracted from heat-inactivated SARS-CoV-2 virus (BEI NR-52347) with known

**Fig. 2 Sensitivity of MORIARTY in amplification-free setting for detection of SARS-CoV-2. a** Workflow of MORIARTY detection of SARS-CoV-2 through a multiplex one-pot reaction strategy and schematic SARS-CoV-2 genome harboring Orf1ab, S, E, M, and N genes. The spacers of crRNA for three LlCsm complexes, LlCsm_S$_0$_D30A, LlCsm_S$_7$_D30A, and LlCsm_S$_8$_D30A, were reprogrammed to detect SARS-CoV-2 S gene at three indicated locations. **b** Serially diluted transcript of S mRNA (S_IVT_RNA) at 0.5 fM-50 fM (12.5 μL) were used to stimulate a 25 μL MORIARTY reaction containing LlCsm_S$_0$_D30A, LlCsm_S$_7$_D30A, and LlCsm_S$_8$_D30A at 250 nM each, 250 nM LlCsm6, 33 mM Tris acetate pH 7.6 at 32 °C, 66 mM potassium acetate, 2.0 μM RNA-FAM, 10 mM MgCl$_2$, and 1.5 mM ATP. The slope of fluorescence rise (right) was determined by employing simple linear regression to the data obtained from the initial 30 min of the experiment performed in triplicates ($n = 3$). All the slopes were compared to the negative control by performing Ordinary one-way analysis of variance (ANOVA) and Dunnett's multiple comparisons test. ****$p < 0.0001$, ***$p = 0.0001$, and **$p = 0.0035$. **c** The independently quantified SARS-CoV-2 control RNA samples (BEI NR-52347) (12.5 μL) were serially diluted to $10 \times 10^3$, $5 \times 10^3$, $2 \times 10^3$ cp/μL and used to stimulate the same 25 μL MORIARTY reaction as in **b**. The slope of fluorescence rise (right) was determined by employing simple linear regression to the data obtained from the initial 30 min of the experiment performed in triplicates ($n = 3$). All the slopes were compared to the negative control by performing Ordinary one-way analysis of variance (ANOVA) and Dunnett's multiple comparisons test. ****$p < 0.0001$, ***$p = 0.0018$. **d** The RNA samples (12.5 μL) extracted from nasopharangeal swabs of six patients or water (5 positive and 1 negative by qRT-PCR) were used to stimulate the same 25 μL MORIARTY reaction as in **b**. The slope of fluorescence rise (right) was determined by employing simple linear regression to the data obtained from the initial 30 min of the experiment performed in triplicates ($n = 3$). All the slopes were compared to the negative control by performing Ordinary one-way analysis of variance (ANOVA) and Dunnett's multiple comparisons test. ****$p < 0.0001$, **$p = 0.0070$ and "ns" denotes non-significant.

viral titers (50,000 cp/μL). Initial experiments suggested that dilutions constituting to ~100 copies/μL could be reliably detected by MORIARTY when RNA-FAM was at 1.5 μM (Supplementary Fig. 3c). Satisfactorily, the RPA primers specific for the S gene lacking T7 promoter sequence did not produce any fluorescence signal as expected (Supplementary Fig. 3b and Supplementary Table 5). Interestingly, unlike amplification-free MORIARTY, addition of a small amount of Mn$^{2+}$ (0.5 mM) and in presence of both RNA-FAM and DNA-FAM had a noticeable benefit when the target DNA concentration was low (Supplementary Fig. 3c). The effect of Mn$^{2+}$ seems to be through suppressing the water signal (Supplementary Fig. 3c). At the Mg$^{2+}$/Mn$^{2+}$ condition and with both RNA-FAM and DNA-FAM probes, we performed serial dilutions from the stock in a range of $500-31$ cp/μl in water and used them in triplicate RT-RPA reactions. Samples with copy numbers higher than 62 cp/μL produced statistically significant rise above water in multi-replicate MORIARTY assays (Fig. 3b).

Finally, we applied RT-RPA MORIARTY to human patient samples. The extracted RNA from additional fourteen patients was split for simultaneous detection with MORIARTY and the q-RT-PCR procedure, respectively (Fig. 3a). Since the q-RT-PCR method detected four targets, human RNase P, SARS-CoV-2 N1, N2 and E genes (Supplementary Table 6), while MORIARTY detected S gene in each sample (Supplementary Table 3), we plotted the cycle threshold (Ct values) of all three viral targets of each patient sample. For MORIARTY results, we plotted the slope of the fluorescence progress of the same patient (Fig. 3c). In general, we observed an overall good agreement between MORIARTY results with the detected E gene titer with q-RT-PCR (Fig. 3c). MORIARTY correctly diagnosed three of the four q-RT-PCR E gene-diagnosed negative and seven of the eight q-RT-PCR E gene-diagnosed positive patients with excellent statistics (Fig. 3c and Supplementary Table 7). MORIARTY also only yielded background-level signal for the one sample yielded no PCR signal for all targets including RNase P (Fig. 3c). The agreement can be extended to MORIARTY results using the end-point fluorescence intensities (Supplementary Fig. 4). Interestingly, the q-RT-PCR results for the N1 and N2 genes agreed with those of MORIARTY to a less degree, likely reflecting the difference in target site accessibility or annealing thermodynamics between the N and the S gene primers.

## Discussion

We have constructed and demonstrated the effectiveness of a Type III-A CRISPR-Cas system for virus detection employing both amplification-free and RT-RPA based strategies. Similar to other CRISPR-Cas-based virus detection methods, MORIARTY-based detection is a highly sensitive system and has advantage

over those based on qPCR without the requirement for thermocyclers[46]. Furthermore, MORIARTY may be easily incorporated with the already developed point-of-care devices/technologies[47]. Unique to MORIARTY is the inherent signal amplification through the RNA-induced cyclic oligoadenylate production. MORIARTY is also unique in that it exploits multiprongness, which yields cumulative signals for detection under multiple buffer conditions. A recently described virus detection method based on Type III-A systems offers another unique readout strategy, in addition to cyclic oligoadenylate-activated Csm6 RNase, by using the pyrophosphate and proton generated by Csm1[48]. Similar to the Cas13-derived methods[20], the Csm-derived method has a minimal restriction on the target RNA that only requires a non-complementary 8-nt sequence to the 5′-tag of the crRNA. In SARS-CoV-2 genome, no such 8-nt sequence motif is found and furthermore, and if the 8-nt sequence is relaxed to the central 4-nt, there are 96 sites throughout the entire genome that represents <1% of the genome. Also similar to Cas13[26], Csm is sensitive to double mismatches, which may be harnessed in identification of single-nucleotide polymorphisms if a synthetic mismatch is programmed[21]. In contrast, the Cas12-derived methods have limitations on the locations of the target sites due to the requirement for the Protospacer Adjacent Motif (PAM). However, Cas12 is able to distinguish single-base variants[49]. Finally, unlike Cas13 or Cas12 where activator cleavage is required for their respective collateral activities[24,26], Csm uses separate active sites for the activator cleavage and, therefore, the collateral activities do not depend on cleavage of the target RNA.

We found that amplification-free detection of target RNA at low concentrations benefited from a Mg$^{2+}$-only condition that primarily reports the RNA-induced cOA$_6$ synthesis activity without the need for the DNA-FAM fluorescence reporter. The Mg$^{2+}$ ion facilitates cOA$_6$ synthesis that then drives the Csm6 RNase activity. This detection strategy differs from the previously developed tandem nuclease strategy where cleavage of linear oligonucleotides by Cas13 activate the Csm6 RNase activity[30]. The Mg$^{2+}$ ion was also helpful in multiplex detection likely due to its additional roles in stabilizing viral RNA structures. A combination of Mg$^{2+}$ and Mn$^{2+}$, on the other hand, was found to be optimal for the RT-RPA MORIARTY where both cOA$_6$ synthesis and DNase activities were used in reporting the presence of a cognate target. The choice of Mg$^{2+}$ and Mn$^{2+}$ or other possible metal ions, however, may depend on the range of target RNA to be detected and other enzymatic processes present in the detection reaction.

ATP in general increases sensitivity because it increases the availability of cOA$_6$. However, the amount of ATP used depends

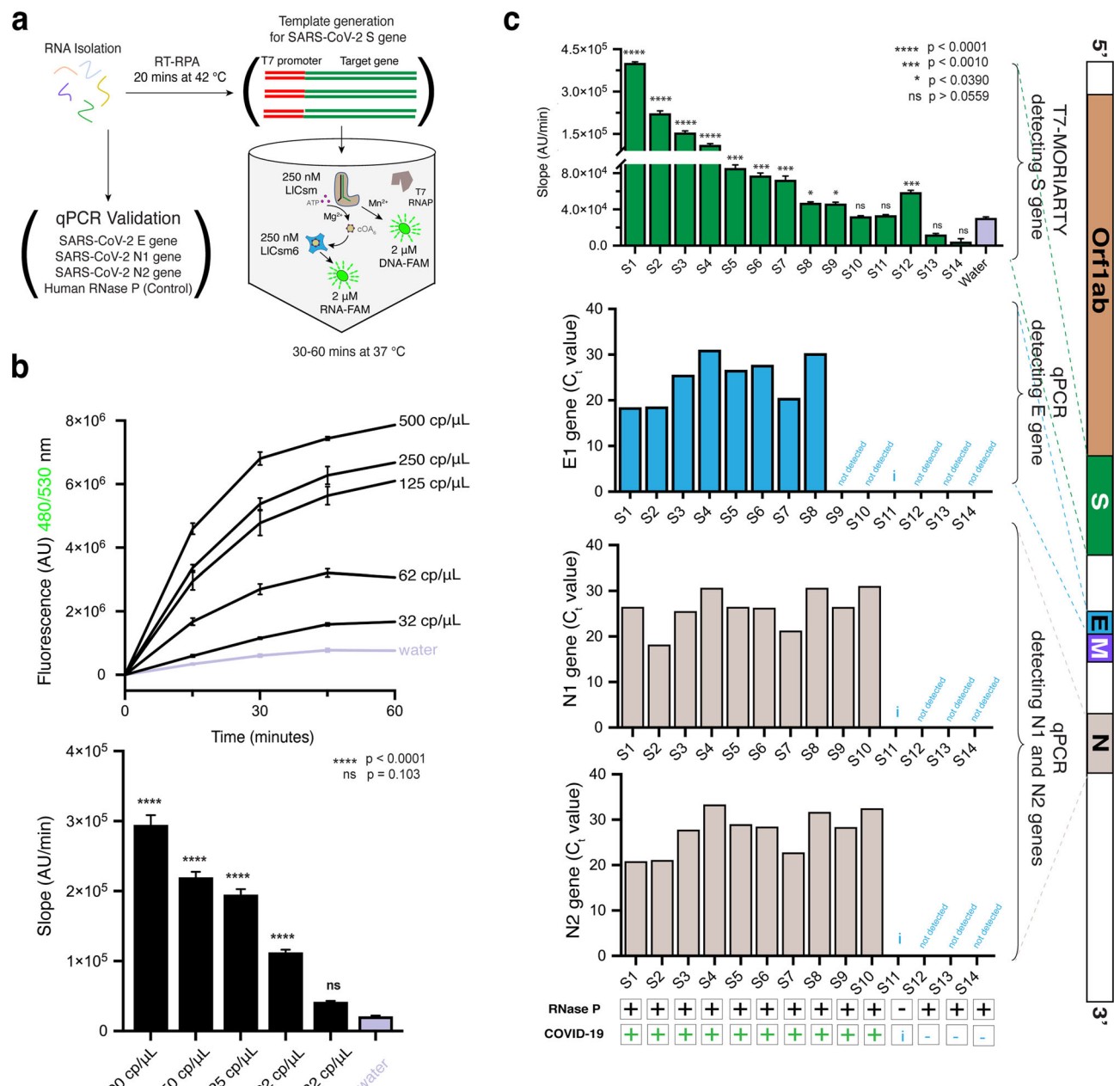

**Fig. 3 Detection of SARS-CoV-2 by RT-RPA and T7-MORIARTY in patient samples. a** Workflow of the RT-RPA and T7-MORIARTY using the LlCsm_S/ LlCsm6 system. All reactions contain 30 mM K-HEPES pH 7.6, 2 mM Spermidine, 0.01% Triton X-100, 17 mM MgCl₂, 0.5 mM MnCl₂, 2.0 μM DNA-FAM), 2.0 μM RNA-FAM, 250 nM LlCsm_S₀_D30A, 250 nM LlCsm6, 0.5 mM rNTPs, 10 mM TCEP, and 60 μg/mL T7 RNA polymerase. **b** The SARS-CoV-2 control RNA samples (BEI NR-52347) were serially diluted, added to RT-RPA reaction and tested in T7 MORIARTY. As a negative control, RT-RPA treated water was used instead of RT-RPA treated sample. The rise in fluorescence corresponding to 500 cp/μL, 250 cp/μL, 125 cp/μL, 62 cp/μL, and 32 cp/μL relative to water are shown (top). The slope of the curves (bottom) was determined by employing simple linear regression to the data obtained from the initial 20 minutes of the experiment performed in triplicates. All the slopes were compared to the negative control by performing Ordinary one-way analysis of variance (ANOVA) and Dunnett's multiple comparisons test. ****p < 0.0001 and "ns" denotes non-significant. **c** The viral RNA samples extracted from 14 patients were RT-RPA treated and tested in T7-MORIARTY. Note the patient samples S1-S14 differ from those used in Fig. 2. T7-MORIARTY results were obtained as in **b** from the initial 40 minutes of the experiment and depicted as a bar diagram with statistical significance values. The qPCR results were plotted as bar diagrams for the three targets (E, N1, and N2) as the Cₜ values. The positive and the negative detection of RNase P in patient samples are indicated by + and − respectively. The qPCR based positive and the negative diagnosis of COVID-19 in patient samples are indicated by + and − respectively. Patient sample with undetectable human RNase P were invalidated, indicated as i (insufficient to detect).

on which metal ion is present in the reaction. For the Mg²⁺ conditions, a range of 1–2 mM could be used while for the Mn²⁺ conditions, 0.03–0.06 mM was used. Interesting, signal contrast falls off when ATP increases beyond these ranges. Multiple explanations for this phenomenon are possible that include a

negative impact of ATP on the DNase activity or inhibition of fluorescence probe binding by ATP.

We explored amplification-free detection with MORIARTY under a condition containing Mg²⁺ and high ATP concentration and were able to directly detect in vitro transcribed

SARS-CoV-2 S mRNA at ~5 fM with LlCsm targeting a single site. This sensitivity is comparable with or better than the previously reported sensitivities of 10 fM (~6000 copies/μL) by LbuCas13a[50] and 50 fM (~30000 copies/μL) by LwCas13a[21] in amplification-free settings but is insufficient to detect viruses present in nasal swab, serum or urine samples that are in the sub-attomolar range. We thus applied a multiplexing detection strategy against S gene mRNA that reached ~2000 copies/μL detection sensitivity with the SARS-CoV-2 RNA and can identify infected patients with high viral titers within 30 min (50 min in total including RT-RPA step). A similar but more extensive multiplexing detection strategy with Cas13 reached the remarkable sensitivity of sub-attomolar with amplification-free detection[51], indicating the possibility for MORIARTY to achieve the same sensitivity when targeting sites are further expanded.

We took advantage of the enzymatic property of Csm in its activation by RNA transcription by constructing a one-pot RT-RPA-coupled detection of viral RNA. This convenient procedure reached high sensitivity with either model SARS-CoV-2 virus or human patient samples. Strikingly, MORIARTY-based assay results show a consistent detection sensitivity to those by q-RT-PCR and seems to have a larger dynamic range than q-RT-PCR in detecting viral RNA. Noteworthy, the temperature range requirement of 37 °C to 42 °C for all the steps in T7-MORIARTY-based detection eliminates the requirement of expensive equipment and makes it potentially compatible with the low-cost hand warmer-mediated heating solution as demonstrated previously[52].

The Csm-derived MORIARTY is thus a versatile virus detection method with high sensitivity.

Given that the genomes of infectious pathogens are made up of either DNA or RNA, multipronged diagnostic tools such as MORIARTY can be employed towards the detection of multiple nucleic acid targets. With the availability of many known Type III-A CRISPR-Cas systems and their individual biochemical differences, MORIARTY offers a broad range of nucleic acid detections under a wide range of conditions.

## Methods

**Cloning**. The pACYC *Lactococcus lactis* Csm (LlCsm) effector module plasmid encoding Cas6, Csm1-6, and CRISPR locus was as described previously[35,53]. The desired mutations including Csm1 H13A/D14N, Csm3 D30A, and Csm6 R355A were introduced by Q5 mutagenesis (New England Biolabs). To make it convenient for incorporating guide RNA, the guide region was incorporated with two BbsI sites[54]. The plasmids encoding LlCsm-S$_0$ effector complex targeting the S gene of SARS-CoV-2 with 29mer gene-complementarity to nucleotides 22280–22308, LlCsm_S$_7$ targeting nucleotides 24702–24730, and LlCsm_S$_8$ targeting nucleotides 25061–25089 of NCBI MT801051.1 were constructed but either Q5 mutagenesis or through BbsI cloning. All three effectors also harbor the Csm1 H13A and Csm3 D30A mutations. All clones were verified using sequencing primers (Eurofins Genomics).

**Protein expression and purification**. The LlCsm effector complexes were all expressed and purified as described previously[35]. Briefly, The all-in-one pACYC plasmid (Fig. 1a) was transformed to *Escherichia coli* NiCo21(DE3) stain (New England BioLab) and the cells were grown to log phase before induction by addition of 0.3 mM isopropyl β-D-1 thiogalactopyranoside (IPTG). The N-terminal His$_6$-tag on LlCsm2 enabled isolation of LlCsm RNP using Ni-NTA affinity chromatography. The Ni-NTA elution pools were loaded on to a size-exclusion column equilibrated with the storage buffer that contains 20 mM HEPES pH 7.5, 200 mM NaCl, 5 mM MgCl$_2$, 14 mM 2-mercaptoethanol. Each LlCsm complex was concentrated to ~55 μM and stored at −80 °C. As LlCsm6 is not a component of the effector complex, it was produced separately from the pLlCsm6 plasmid (Fig. 1a) by the similar Ni-NTA affinity and gel filtration chromatography[35] and stored in 20 mM HEPES pH 7.5, 300 mM NaCl, 14 mM 2-mercaptoethanol. The T7 RNA polymerase was produced as described previously[54].

**In vitro transcription of S gene mRNA**. The plasmid encoding for the SARS-CoV-2 surface glycoprotein (Spike protein), pGBW-m4046887 (Addgene), was amplified in DH5α cell and subjected to plasmid DNA extraction. The extracted plasmid DNA digested by BamH1 (New England Biolabs) overnight and verified by gel analysis. To produce S mRNA, 1 μg of the linearized DNA was used as the template for in vitro transcription in a reaction containing 1x transcription buffer (10x consists of 500 mM Tris pH 8.0, 100 mM DTT 200 mM MgCl$_2$), 5 mM NTPs, and 480 μg/mL T7 RNA polymerase. The transcription reaction was incubated at 37 °C for 3 h and treated with 2 units of Turbo DNase (Invitrogen). The RNA transcript was purified using Monarch RNA Cleanup kit (New England Biolabs), eluted in water, aliquoted, flash frozen using liquid nitrogen and stored at −80 °C.

**Amplification-free MORIARTY**. For two-channel fluorescence experiments, a short 8-nt DNA-oligo and a short 5-nt RNA-oligo were designed such that the DNA-oligo was covalently linked to 5′ Alexa Fluor 594 (NHS Ester) fluorescent dye and 3′ Iowa Black RQ quencher (DNA-Alexa) while the RNA-oligo was covalently linked to 5′ 6-FAM (Fluorescein) fluorescent dye and 3′ Iowa Black FQ quencher (RNA-FAM) (Supplementary Table 4). The dual fluorescence was simultaneously measured on Spectramax iD5 multi-mode microplate reader (Molecular Devices) using 480 nm/530 nm (to track cleavage of RNA-FAM) and 570 nm/630 nm (to track cleavage of DNA-Alexa) excitation/emission wavelength at 1 min intervals with the SoftMax Pro 7 software. The reactions were performed in triplicates and averaged for the final plots (GraphPad Prism). For single channel experiment, RNA-FAM or RNA-FAM plus DNA-FAM was used and the fluorescence signal was measured using 480 nm/530 nm excitation/emission wavelength at 1 min intervals. The reactions were performed in 0.1-1x TAPA buffer (10x consists of 330 mM Tris acetate pH 7.6 at 32 °C, 660 mM Potassium acetate) containing 0.5–2 μM DNA-Alexa, 0.5–2 μM RNA-FAM, 250 nM LlCsm effector complex, 1–250 nM LlCsm6, 0–10 mM MgCl$_2$/MnCl$_2$, 0–1.5 mM ATP. and 5 fM–500 nM target RNA (or water) at 37 °C. For multiplexing experiments, 250 nM of each LlCsm RNP was included and the final buffer contained 0.1x TAPA, 10 mM MgCl$_2$, and 2 μM RNA-FAM. When the target RNA was either SARS-CoV-2 control or nasal swab extracted patient samples, 12.5 μL RNA at the specified concentrations was added to the 12.5 μL master mix.

**T7-MORIARTY**. The T7-MORIARTY methodology was designed to track LlCsm co-transcriptional activation of template viral DNA obtained either from synthetic DNA (Supplementary Table 5) or from RT-RPA step. All T7-MORIARTY reactions were done using RNA-FAM and DNA-FAM fluorescent probes at a reaction volume of 25 μL or 100 μL as required. The reactions were performed in a buffer cocktail of 1x TAPA buffer (10x consists of 330 mM Tris acetate pH 7.6 at 32 °C, 660 mM Potassium acetate) and 1x transcription buffer (10x contains 300 mM K-HEPES pH 7.6, 20 mM Spermidine, 0.1% Triton X-100, 170 mM MgCl$_2$) containing 2 μM DNA-FAM, 2 μM RNA-FAM, 250 nM LlCsm effector complex, 250 nM LlCsm6, 10 mM MgCl$_2$, 0.5 mM MnCl$_2$, 0.5 mM rNTPs, 10 mM TCEP, 60 μg/mL T7 RNA polymerase, and 0–5 nM target DNA (or 12.5 μL of RT-RPAed product) at 37 °C. The fluorescence was measured on Spectramax iD5 multi-mode microplate reader (Molecular Devices) using 480 nm/530 nm excitation/emission wavelength at 5 min intervals. The reactions were performed in triplicates and averaged for the final plots (GraphPad Prism). The reaction products were transferred to 0.2 mL PCR tubes and imaged against Fluorescein wavelength using Chemidoc imaging system (BioRad).

**RT-RPA**. The RT-RPA was performed using TwistAmp Basic kit following manufacturer's instructions (TwistDx). For each lyophilized pellet provided by the kit, a 50 μL reaction mix containing 29.5 μL of rehydration buffer (TwistAmp Basic kit), 0.5 μM each of forward and reverse primers (Eurofins Genomics) and 100 U Protoscript reverse transcriptase (NEB) was prepared. The master-mix was added to the RPA tube on ice to resuspend the pellet. After the pellet was dissolved completely, 5 μL RNA samples extracted from patient nasopharyngeal swab (Zymo Research) using QIAamp viral mini kit (Qiagen) was added. To initiate the RT-RPA reaction, 14 mM Magnesium Acetate (provided with the kit) was added and the reactions were incubated for 30 min at 42 °C with intermittent mixing every 10 min. After the reaction was complete, the RT-RPA product was transferred to ice until T7-MORIARTY detection assay was ready.

**q-RT-PCR**. To quantify copy numbers of in vitro transcribed S gene mRNA (S_IVT_RNA), we performed a 2-step laboratory-based assay in ABI 7500 Fast Real-Time PCR System. Briefly, cDNA was synthesized using SuperScript III First-Strand cDNA kit (Invitrogen) followed by DNA amplification using PerfeCTa SYBR Green FastMix (Quantabio) and S gene specific RPA primers (Supplementary Table 6) under a standard PCR cycle condition. The heat-inactivated SARS-related Coronavirus 2, isolate USA-WA1/2020 (BEI NR-52347) with serial dilutions was used as the copy standards.

To detect patient samples via RT-PCR, we performed the assay consistent with the FDA EUA "OSCEOLA SARS-CoV-2 Real-time Reverse Transcriptase (RT)-PCR Diagnostic Assay." Briefly, 5 μL RNA samples extracted from patient nasopharyngeal swab (Zymo Research) using QIAamp viral mini kit (Qiagen) was used in each of four independent qPCR reactions using the CDC primer-probe sets for human RNase P, chr10:90872001-90872065 (hg38), SARS-CoV-2 Nucleocapsid target 1, 2787-28332 (NC_045512), SARS-CoV-2 Nucleocapsid target 2 29164-29210 (NC_045512), and the WHO primer-probe sets for human SARS-CoV-2

Envelope, 26294-26357 (NC_045512), (Supplementary Table 6). 20 μL reactions were prepared using the Luna Universal Probe One-Step RT-qPCR Kit (New England Biolabs, #E3006) and were run on an Analytik Jena qTower3 84 Real-Time PCR System with 21 CFR Part 11 software module-enabled qPCRsoft384 software version 1.2.3.0 (Analytik Jena), under the following cycling conditions: Reverse Transcription at 55 °C for 10 min, Initial Denaturation at 95 °C for 1 min, followed by 40 cycles of Denaturation at 95 °C for 10 s, and Extension at 60 °C for 30 s. Cycle threshold ($C_t$) values of less than 35 are reported as "positive" for that measurand, while a $C_t$ of <35.00, or "No $C_t$" is reported as "negative."

**Reporting summary**. Further information on research design is available in the Nature Research Reporting Summary linked to this article.

## Data availability

The raw fluorescence intensity readings generated in this study are provided in the Source Data file. The cycle threshold values generated in this study are provided in Supplementary Tables 7 and 8. Source data are provided with this paper.

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

## Acknowledgements

The authors wish to thank FSU Cloning and Sequencing facilities, Michael P. Terns (University of Georgia) for the original LlCsm and the LlCsm6 expression plasmids. This work was supported by NIH Grant R01 GM099604 to H.L.

## Author contributions

S.S., C.W., and H.L. developed the initial experimental protocol. S.S. designed the employed nucleic acid targets and guides and expressed and purified LlCsm effector complexes with the assistance of C.W. and H.N.G.; C.W. expressed and purified LlCsm6 protein; S.S. and H.N.G. performed and optimized the detection reactions, analyzed the data, and drafted the paper. J.H.D provided COVID patient samples and performed RT-qPCR experiments. S.S., H.N.G., and H.L. made the figures. All authors edited the manuscript.

## Competing interests

The authors declare no competing interests.
