## [Peer Review File · Nature Communications]

Reviewers' Comments:

Reviewer #1:

Remarks to the Author:

To combat the global viral pandemics, there is an urgent need for cheap, fast and reliable point of care testing technologies for virus detection. Class 2 CRISPR-Cas systems that are defined by the presence of a single-protein effector, as exemplified by Cas9, Cas12 and Cas13, have been already employed for diagnostic applications. In the current manuscript Sridhara et al adopted Class 1 type III-A CRISPR-Cas system for SARS-CoV-2 detection. Effector complexes of Type III systems are comprised of multiple subunits and exhibit diverse catalytic activities. Sridhara et al combined ssDNAse activity of type-III A CRISPR-Cas system from *Lactococcus lactis* with a cOA-activated collateral LICsm6 RNase activity to develop MORIARTY detection platform that was employed both for amplification-free and RT-RPA amplification-dependent detection of SARS-CoV-2 virus. Authors claim that they were able to reach 3000 copies/uL sensitivity in amplification-free and 62 copies/uL sensitivity if RT-RPA-amplification step is included.

Major concerns:

1) For viral RNA detection, Sridhara et al propose to monitor combined ssDNAse (Csm1) and ssRNAse (Csm6) activities using fluorescently labelled nucleic acids. To this aim, authors established optimal reaction conditions (ATP, metal-ion cofactor concentrations) for both ssDNAse (Csm1) and ssRNAse (Csm6) activities. Under these conditions the overall fluorescence yield is a resultant of i) DNA degradation by the Csm1, ii) RNA degradation by the Csm6 RNase that is controlled by the rate of cOA6 synthesis by Csm1 protein and cOA6 degradation by the CARF domain of Csm6 RNase.

cOA6 synthesis that is triggered in response to viral RNA binding by the Csm complex is ATP dependent. Low ATP concentrations used in the fluorescent cleavage assay may result in low cOAn yields and different range of cOAn molecules. What cOAn (n=?) molecules are predominant under the nuclease assay conditions? What are rates for cOA degradation by CARF domain of Csm6 RNase?

2) In contrast to type I and II CRISPR-Cas systems, the type III system often show a relaxed specificity that manifests in mismatched RNA target binding by the Csm complex. How specific is MORIARTY detection system? The fluorescence signals produced by the cognate RNA, different mismatched RNAs and non-specific RNA have to be compared.

3) LICsm effector complex shows collateral DNA cleavage even in the absence of target RNA. How does this affect the fluorescence signal and specificity of MORIARTY system? To address this point, in vitro analysis of LICsm complex DNase activity in the absence of target RNA should be performed and fluorescence signal in the absence of target RNA should be compared with a fluorescence signal observed in presence of target RNA.

4) Data presented in Figure S1 for LICsm-HD/LICsm6-R365A (+10mM MgCl₂, +0,5mM ATP) contradicts with data for LICsm-WT/LICsm6-R365A ((+10mM MgCl₂, +0,5mM ATP). Why there is no RNase activity observed for LICsm-WT/LICsm6-R365A? The observed increase of FAM-RNA signal for LICsm-HD/LICsm6-R365A mutant looks like a "false positive" signal and therefore raises the question on method reliability. Authors speculate that the ssRNAse activity of Csm1 previously reported in their NAR paper (reference [40] is wrong) may be responsible for RNA degradation. However, the activity reported in NAR paper is i) for a different system; ii) SeCsm1 HDmut generated a similar RNase cleavage pattern and showed similar (or even less) activity as the wild-type SeCsm1 (Figure 6C). Moreover, SeCsm1 RNase activity reported in the NAR paper is for an isolated SeCsm1 protein, but not for the entire Csm complex. Therefore, additional data are required to explain experimental result presented in Figure S1 (LICsm-HD/LICsm6-R365A,+10mM MgCl₂, +0,5mM ATP)

5) In MORIARTY augmentability experiment, which is presented in Figure1B (Row 2, middle panel, 10mM MnCl₂, 0,05mM ATP), the signal increase observed after addition DNA-FAM to the RNA-FAM substrate is higher compared to the signal when RNA-FAM was added to the DNA-FAM. How this could be explained?

6) Since Mg²⁺ supports only cOA-synthase activity of LICsm complex but not DNase activity, the usage of two substrates (see Figure 1B, the experiment with RNA-FAM and DNA-FAM, 10 mM Mg, 0.5 mM ATP) results in a negligible signal increase compared to the one when a single RNA or DNA substrates are used. Therefore, no conclusions on the cumulative effect could be done and this Figure should be moved to Supplementary Data.

Same is valid for the experiment presented in Figure 1C (experiment with RNA-FAM and DNA-FAM, 17 mM Mg, 0.5 mM ATP, 0mM MnCl₂). In the absence of Mn²⁺, there is no signal from LICsm DNase activity, therefore this experiment does not demonstrate the augmentability of T7-MORIARTY. This Figure should be removed or moved to Supplementary Data.

7) The discussion would benefit if MORIARTY detection system is more broadly compared to other published CRISPR-Cas diagnostic platforms (SATORI, DETECTR and others) including both pros and cons.

Minor:

1) Incorrect citation [40].

2) The same green color is used for two different curves corresponding 1uM RNA-FAM and 1 DNA-FAM in the legend of Figures 1B and 1C. This looks confusing.

4) p.9: "Initial experiments suggested that dilutions constituting to ~100 copies/uL could be reliably detected by MORIARTY (Supplementary Figure S4)." The curve corresponding 100copies/uL looks nearly identical to control curves of water and T7-minus primer.

5) Please indicate exact experimental conditions used in Figures 2B and 2C, 3B and 3C

6) p.11, correct MOARTY to MORIARTY

Reviewer #2:

Remarks to the Author:

Sridhara and colleagues describe the development and testing of the class I type III-A CRISPR-Cas system for detection of specific RNA targets, a method which they named MORIARTY. They determined the optimal conditions for nucleic acid detection and test SARS-CoV-2 RNA detection with RT-RPA on a set of patient samples side-by-side with RT-qPCR. Although the authors present the first application of this entire complex for nucleic acid detection (of note, Csm6 in combination with other Cas systems has been used previously for amplifying detection signals), this manuscript would require substantial changes to the presentation of the results and it is necessary that the results are communicated in the context of the many other CRISPR-based detection technologies and in particular those for SARS-CoV-2 detection. I have outlined specific concerns below:

Major concerns:

1. The introduction is lacking adequate references to the substantial number of published papers describing the use of CRISPR systems for detecting viral RNA and in the prior year SARS-CoV-2 detection technologies. By failing to discuss these technologies, it is hard to evaluate the true benefits of this method over others – sensitivity? sample-to-answer time? ease of implementation? cost (particularly because a multicomponent complex is required)? Compatibility with visual readouts?

2. Related to the above point, the authors should discuss throughout the results and the discussion about how the optimization of their method either improved sensitivity or speed of the assay as these are two key metrics in the evaluation of detection technologies.

3. It is difficult to determine the reliability and consistency of much of the data presented because most figure panels do not have error bars. The authors describe that measurements were performed in triplicate but what is the spread of these values. The number of replicates should be included in the figure captions.

4. In Figures 2 and 3, negative controls such as water have been included. However, this essential control was not included in any of the optimization experiments. This is a key control because some conditions could have lower background signal and could improve the method's ability to differentiate between lower input conditions and negative samples.

5. The use of many subpanels within individual figure panels (e.g. Figure 1B and 1C) make early portions of the results section extremely difficult to follow. The main text figures also include tons of optimization, and I would suggest moving lesser points to supplemental figures or alternatively summarizing the data using other visualization approaches such as heatmaps.

6. I applaud the authors for testing their method on patient samples and for testing a set of patient samples at higher Ct values (lower viral quantities). However, I would present non-normalized MORIARTY and Ct values for the patient samples. RNase P is typically used as an internal control to ensure that RNA was appropriately extracted from the sample and there isn't a relationship between viral RNA levels and RNase P and therefore values should not be normalized in this way.

7. It would also be helpful to show how translatable this approach is to other targets. How flexible is the method? Do you see similar sensitivities for different target RNAs or regions of SARS-CoV-2? I would hesitate to make broad claims about this method's use for viral detection when it is the first time this complex has been used for this application and only one target is tested.

Minor concerns:

1. Please include line numbers in submissions as it makes it easier to reference small suggestions within all parts of the text.

2. Introduction paragraph 1: It is too strong to say as a broad statement that human infectious diseases are highly contagious – some are not but not all.

3. Introduction paragraph 1: "thereby limiting" -> which could limit

4. Introduction paragraph 2: there are several inaccurate statements: (a) antigen detection can distinguish between active and previous infection because these tests detect viral protein which exists during active infection. I think the authors are mischaracterizing this technology with serology testing which cannot distinguish between the two and serology tests are only valid when antibodies have been elicited. (b) nucleic acid detection is not a recent development. PCR testing has been the gold-standard for pathogen detection for quite a long time. (c) it is inaccurate to describe CRISPR detection systems to be at the "home-brew" stage; many still require reagents with cold-chain and pipetting steps.

5. Figure 1A: the representation of the active and inactive RNAs in the context of the Csm complex is confusing. Is the 5' tag a separate RNA or is it that binding of the extended region leads to inactivity as the NTR has been depicted. Also, it is unclear why the Csm3 RNase activity has been highlighted in this figure if none of the reporters read out this activity.

6. End of results paragraph 2, sentence "These results show that detection of viral RNA by MORIARTY can be carried out in multiple buffer conditions." This sentence overstates the results: the target RNA used for these experiments has not been described as virally derived and only one concentration of target RNA has been tested. Also, it appears that multiple buffer conditions are possible, but only if Mn is present.

7. Switch Figure 1B top right with Figure S1 middle row middle panel so that ATP concentrations are consistent throughout top row of Figure 1B.

8. Figure 1C, top left panel: please clarify in the figure itself if this is the condition without the T7 promoter resulting in no signal is observed for either reporter.

9. Figure captions should not include results interpretation; see Figure 1C caption as an example.

10. Figure S2: comparison between conditions in a single row would be simpler if data were plotted on the same graph with each ATP concentration as a different color.

11. SARS-CoV-2 RNA design: it would be helpful to discuss the level of sequence conservation at this site and whether given the sequence constraints of this system whether there are other sites that could be selected in case mutations accumulate in this location.

12. Discussion section: it is incorrect to reference the sensitivity of SHERLOCK as 50 fM. The sensitivity of Cas13 detection was observed to be this, but the SHERLOCK method includes both amplification and Cas13-based detection. I would also caution comparing MORIARTY's sensitivity

to these earlier publications as target RNAs were different and there are more recent publications that could serve as a more relevant comparator.

13. The authors should test or add discussion as to whether or not their amplification-free method would detect any of the patient samples in order to convey in what contexts this amplification-free method could be used.

Responses to REVIEWER COMMENTS

Reviewer #1 (Remarks to the Author):

To combat the global viral pandemics, there is an urgent need for cheap, fast and reliable point of care testing technologies for virus detection. Class 2 CRISPR-Cas systems that are defined by the presence of a single-protein effector, as exemplified by Cas9, Cas12 and Cas13, have been already employed for diagnostic applications. In the current manuscript Sridhara et al adopted Class 1 type III-A CRISPR-Cas system for SARS-CoV-2 detection. Effector complexes of Type III systems are comprised of multiple subunits and exhibit diverse catalytic activities. Sridhara et al combined ssDNAse activity of type-III A CRISPR-Cas system from *Lactococcus lactis* with a cOA-activated collateral LICsm6 RNase activity to develop MORIARTY detection platform that was employed both for amplification-free and RT-RPA amplification-dependent detection of SARS-CoV-2 virus. Authors claim that they were able to reach 3000 copies/uL sensitivity in amplification-free and 62 copies/uL sensitivity if RT-RPA-amplification step is included.

Major concerns:

1) For viral RNA detection, Sridhara et al propose to monitor combined ssDNAse (Csm1) and ssRNAse (Csm6) activities using fluorescently labelled nucleic acids. To this aim, authors established optimal reaction conditions (ATP, metal-ion cofactor concentrations) for both ssDNAse (Csm1) and ssRNAse (Csm6) activities. Under these conditions the overall fluorescence yield is a resultant of i) DNA degradation by the Csm1, ii) RNA degradation by the Csm6 RNase that is controlled by the rate of cOA6 synthesis by Csm1 protein and cOA6 degradation by the CARF domain of Csm6 RNase.

cOA6 synthesis that is triggered in response to viral RNA binding by the Csm complex is ATP dependent. Low ATP concentrations used in the fluorescent cleavage assay may result in low cOAn yields and different range of cOAn molecules. What cOAn (n=?) molecules are predominant under the nuclease assay conditions? What are rates for cOA degradation by CARF domain of Csm6 RNase?

Response: The reviewer raised two interesting questions previously not answered for LICsm6. The first is which one of the cOAn molecules LICsm produces. The second is (if LICsm6 degrades cOAn), what is the rate of this activation. We now have more evidence for addressing both questions.

We have three pieces of evidence supporting that LICsm produces cOA₆. 1) In reference 34, we showed by thin layer chromatography and radioactively labeled ATP that an overnight reaction of LICsm produced a high molecular weight cyclicoligoadenylate that did not further polymerize; 2) we now performed mass spectroscopy analysis of this reaction product and showed that LICsm predominately produces cOA₆; 3) we also performed RNA cleavage by LICsm6 in presence of synthetic cOA₃, cOA₄ and cOA₆, which showed consistently that LICsm6 only cleaved RNA in the presence of cOA₆ and not cOA₃ or cOA₄. Results of 2) and 3) will be included in revised reference 34 and here for reviewer's assessment (next page).

A LICsm produces primarily cOA₆

B LICsm6 cleaves RNA in response to cOA₆ but not cOA₃ or cOA₄

With regard to cOA degradation, we also obtained evidence that LICsm6 does degrade cOA₆ by a new fluorescence experiment where a duplicated LICsm reactions were either pretreated with 50 nM Csm6 or buffer for 30 minutes followed by addition of fluorescence probe or probe plus Csm6. We indeed observed lowered fluorescence signal in the Csm6-pretreated reaction, suggesting some of the cOA₆ produced were degraded by Csm6. We now include this data in Supplementary Figure 2A. As to the rate of degradation, we believe that it requires additional careful kinetics experiments that could delay this work.

However, we estimate it is the range of minutes, although we only performed this experiment at one LICsm6 concentration.

We thank reviewer for this request because this result may lead to improved detection sensitivity if we are able to prevent cOA₆ degradation without affecting Csm6's ability to sensing cOA₆ in MORIARTY.

2) In contrast to type I and II CRISPR-Cas systems, the type III system often show a relaxed specificity that manifests in mismatched RNA target binding by the Csm complex. How specific is MORIARTY detection system? The fluorescence signals produced by the cognate RNA, different mismatched RNAs and non-specific RNA have to be compared.

Response: This is a very good question. We tested a series mismatched targets and were able to access specificity of MORIARTY (Supplementary Figure 3A). As the new supplementary data show that in the T7-MORIARTY mode, although single mismatches in the seed region between the target DNA and guide RNA reduced the fluorescence signal by ~40%, double mismatch could reduce the signal by nearly 90%. This finding is encouraging for us to construct MORIARTY for detection of mutant SARS-CoV-2 variants such as the Delta variant. We included the data and the discussion of its implication.

3) LICsm effector complex shows collateral DNA cleavage even in the absence of target RNA. How does this affect the fluorescence signal and sp^{1,2} specificity of MORIARTY system? To address this point, in vitro analysis of LICsm complex DNase activity in the absence of target RNA should be performed and fluorescence signal in the absence of target RNA should be compared with a fluorescence signal observed in presence of target RNA.

Response: We agree. This is the reason that all assays were compared to the background signals in the absence of target RNA.

4) Data presented in Figure S1 for LICsm-HD/LICsm6-R365A (+10mM MgCl₂, +0,5mM ATP) contradicts with data for LICsm-WT/LICsm6-R365A ((+10mM MgCl₂, +0,5mM ATP). Why there is no RNase activity observed for LICsm-WT/LICsm6-R365A? The observed increase of FAM-RNA signal for LICsm-HD/LICsm6-R365A mutant looks like a “false positive” signal and therefore raises the question on method reliability. Authors speculate that the ssRNase activity of Csm1 previously reported in their NAR paper (reference [40] is wrong) may be responsible for RNA degradation. However, the activity reported in NAR paper is i) for a different system; ii) SeCsm1 HDmut generated a similar RNase cleavage pattern and showed similar (or even less) activity as the wild-type SeCsm1 (Figure 6C). Moreover, SeCsm1 RNase activity reported in the NAR paper is for an isolated SeCsm1 protein, but not for the entire Csm complex. Therefore, additional data are required to explain experimental result presented in Figure S1 (LICsm-HD/LICsm6-R365A,+10mM MgCl₂, +0,5mM ATP)

Response: The reason for no RNase activity with the LICsm WT/LICsm6-R365A combination is because that LICsm6-R365A is an RNA cleavage-deficient mutant (see the RNA cleavage figure on the previous page). However, the reviewer is correct about the suspicious activity of this batch of HD mutant. We re-purified LICsm HD mutant twice and found that it did behave accordingly (i.e., no RNase activity). We believe that the suspicious RNase activity was a result of protein contamination of the previous batch. We corrected this mistake.

5) In MORIARTY augmentability experiment, which is presented in Figure1B (Row 2, middle panel, 10mM MnCl₂, 0,05mM ATP), the signal increase observed after addition DNA-FAM to the RNA-FAM substrate is higher compared to the signal when RNA-FAM was added to the DNA-FAM. How this could be explained?

Response: We believe that our figure might have caused some misunderstanding. Both green curves are independent experiments with either DNA-FAM or RNA-FMA, respectively, thus measuring the signal of either activity independently. The blue curve is when both probes are present in the same reaction (note there is no order in addition), which shows an additive effect (higher signal than either one of them). In revised Figure 1, the similar panel was labeled more clearly.

We also believe the curve with half concentration of the two probes (cyan) is distracting and thus removed it.

6) Since Mg²⁺ supports only cOA-synthase activity of LICsm complex but not DNase activity, the usage of two substrates (see Figure 1B, the experiment with RNA-FAM and DNA-FAM, 10 mm Mg, 0.5 mM ATP) results in a negligible signal increase compared to the one when a single RNA or DNA substrates are used. Therefore, no conclusions on the cumulative effect could done and this Figure should be moved to Supplementary Data.

Same is valid for the experiment presented in Figure 1C (experiment with RNA-FAM and DNA-FAM, 17 mm Mg, 0.5 mM ATP, 0mM MnCl₂). In the absence of Mn²⁺, there is no signal from

LICsm DNase activity, therefore this experiment does not demonstrate the augmentability of T7-MORIARTY. This Figure should be removed or moved to Supplementary Data.

Response: The reviewer is correct that in Mg^{2+} alone, only RNase activity is active. There is no need for us to confirm this concept. We removed the unnecessary panels.

In fact, we more carefully address the augmentation question under the actual testing condition (Supplementary Figure 2). In our new multiplexing and amplification-free detection, we reached high sensitivity without Mn^{2+} . Here, we did not include DNA-FAM as the probe. Our optimization of RT-RPA MORIARTY showed signal augmentation, although small, when LICsm6 and ATP are high and when Mg^{2+}/Mn^{2+} both are present, (Figure 1C and Supplementary Figure 3). In this case, we included both probes (Figure 3). We carefully discussed our rationales for the respective choices in probes.

7) The discussion would benefit if MORIARTY detection system is more broadly compared to other published CRISPR-Cas diagnostic platforms (SATORI, DETECTR and others) including both pros and cons.

Response: We significantly enhanced the discussion (both in Introduction and in Discussion) with respect to other CRISPR diagnostic methods to highlight the unique aspects of MORIARTY among the existing ones.

Minor:

1) Incorrect citation [40].

Response: Since we corrected HD double mutant result, we removed reference 40.

2) The same green color is used for two different curves corresponding 1uM RNA-FAM and 1 DNA-FAM in the legend of Figures 1B and 1C. This looks confusing.

Response: We kept the green color for both DNA-FAM and RNA-FAM as they emit green fluorescence, but we now clearly labeled each curve in the revised Figure 1.

4) p.9: "Initial experiments suggested that dilutions constituting to ~100 copies/uL could be reliably detected by MORIARTY (Supplementary Figure S4)." The curve corresponding 100copies/uL looks nearly identical to control curves of water and T7-minus primer.

Response: We now completely revised the referred figure where we included an experiment with increased probe concentration and saw a clear separation between 100 cp/ul and water signal.

5) Please indicate exact experimental conditions used in Figures 2B and 2C, 3B and 3C

Response: We now added the reaction conditions in figure captions and more carefully edited the method part.

6) p.11, correct MOARITY to MORIARTY

Response: Corrected.

Reviewer #2 (Remarks to the Author):

Sridhara and colleagues describe the development and testing of the class I type III-A CRISPR-Cas system for detection of specific RNA targets, a method which they named MORIARTY. They determined the optimal conditions for nucleic acid detection and test SARS-CoV-2 RNA detection with RT-RPA on a set of patient samples side-by-side with RT-qPCR. Although the authors present the first application of this entire complex for nucleic acid detection (of note, Csm6 in combination with other Cas systems has been used previously for amplifying detection signals), this manuscript would require substantial changes to the presentation of the results and it is necessary that the results are communicated in the context of the many other CRISPR-based detection technologies and in particular those for SARS-CoV-2 detection. I have outlined specific concerns below:

Major concerns:

1. The introduction is lacking adequate references to the substantial number of published papers describing the use of CRISPR systems for detecting viral RNA and in the prior year SARS-CoV-2 detection technologies. By failing to discuss these technologies, it is hard to evaluate the true benefits of this method over others – sensitivity? sample-to-answer time? ease of implementation? cost (particularly because a multicomponent complex is required)? Compatibility with visual readouts?

Response: We revised the Introduction that now clearly describes the invention and major improvement of the Cas12 and Cas13-based method. We also point out the fact that these systems lack internal signal amplifications and the possibility for dual channel detection that are the unique features of MORIARTY.

2. Related to the above point, the authors should discuss throughout the results and the discussion about how the optimization of their method either improved sensitivity or speed of the assay as these are two key metrics in the evaluation of detection technologies.

Response: We have revised the Result section completely and added additional supplementary data to detail the process for us to reach the required sensitivity, and now with amplification-free. The most significant improvement is the elevated cOA_6 synthesis and multiplexing.

3. It is difficult to determine the reliability and consistency of much of the data presented

because most figure panels do not have error bars. The authors describe that measurements were performed in triplicate but what is the spread of these values. The number of replicates should be included in the figure captions.

Response: The reviewer may be referring to Figure 1 or the original Supplementary Figure 1 & 2 where we did not provide replicates, nor we provided background-correction under each condition. We addressed this concern in two ways. First, we completely reperfomed the experiments in Figure 1 and S1 and provided background (water)-corrected results. There is no change in conclusion because these figures were for establishing the concept and performed with very high target concentrations (500 nM) that induce strong fluorescence. Second, Figure S2 is not necessary as Figure S1 reveals the same message and thus removed.

For other experiments when target concentrations are low, especially those critical for determining sensitivity, we carefully performed triplicates and with water (also triplicates) with multi-channel pipette. As reviewer can see, most error bars are very reasonable. For intermediate and optimization trials (for instance, Csm6 or ATP concentrations) such as those shown in revised Supplementary Figure S2 and S3, we did not triplicate them until we finalize the conditions, although all were performed with water under the corresponding conditions.

4. In Figures 2 and 3, negative controls such as water have been included. However, this essential control was not included in any of the optimization experiments. This is a key control because some conditions could have lower background signal and could improve the method's ability to differentiate between lower input conditions and negative samples.

Response: In revised optimization experimental results, all included water at the corresponding conditions.

5. The use of many subpanels within individual figure panels (e.g. Figure 1B and 1C) make early portions of the results section extremely difficult to follow. The main text figures also include tons of optimization, and I would suggest moving lesser points to supplemental figures or alternatively summarizing the data using other visualization approaches such as heatmaps.

Response: We have revamped the figures completely, both making it easier to follow and reflecting our new experiments with multiplexing. We reduce the introductory panels (such as the original Figure S2) and focused on presenting results more relevant to testing (new Figure S2 and S3).

6. I applaud the authors for testing their method on patient samples and for testing a set of patient samples at higher Ct values (lower viral quantities). However, I would present non-normalized MORIARTY and Ct values for the patient samples. RNase P is typically used as an internal control to ensure that RNA was appropriately extracted from the sample and there isn't a relationship between viral RNA levels and RNase P and therefore values should not be

normalized in this way.

Response: We thank the reviewer for this advice. We have now present Figure 3 without normalizing RNase P value. Note that there is no change in conclusion.

7. It would also be helpful to show how translatable this approach is to other targets. How flexible is the method? Do you see similar sensitivities for different target RNAs or regions of SARS-CoV-2? I would hesitate to make broad claims about this method's use for viral detection when it is the first time this complex has been used for this application and only one target is tested.

Response: We thank reviewer for this suggestion. In addition to the original target, S0, we now added two targets on the S-gene of SARS-Cov-2 (S7 and S8) and purified the two RNPs respectively. We showed that when used individually, S7 and S8 RNP both can detect in vitro transcribed S mRNA, although with some differences in sensitivity. The benefit of this exercise is to enable us to detect the virus RNA through multiplexing. When S0, S7, S8 were simultaneously used to target S mRNA in a multiplexing MORIARTY reaction, we reached sensitivity of 0.5 fM for in vitro transcribed S mRNA or ~2000 copies / ul SARS-CoV-2 RNA. Finally, we were able to direct detect patient samples when Ct values were low. The ability of S7 and S8 to be used in MORIARTY demonstrates its general applicability and another revenue to improve sensitivity.

Minor concerns:

1. Please include line numbers in submissions as it makes it easier to reference small suggestions within all parts of the text.

Response: Line numbers are now added.

2. Introduction paragraph 1: It is too strong to say as a broad statement that human infectious diseases are highly contagious – some are not but not all.

Response: We revised the phrase to “Some of these infections are highly contagious”

3. Introduction paragraph 1: “thereby limiting” -> which could limit

Response: corrected.

4. Introduction paragraph 2: there are several inaccurate statements: (a) antigen detection can distinguish between active and previous infection because these tests detect viral protein which exists during active infection. I think the authors are mischaracterizing this technology with serology testing which cannot distinguish between the two and serology tests are only valid when antibodies have been elicited. (b) nucleic acid detection is not a recent development. PCR testing has been the gold-standard for pathogen detection for quite a long time. (c) it is inaccurate to describe CRISPR detection systems to be at the “home-brew” stage;

many still require reagents with cold-chain and pipetting steps.

Response: We see how it was incorrectly phrased with both antigen-based and serological tests mixed. This section is significantly revised to clarify these types of tests.

We also removed the phrase “recent” when referring nucleic acid testing.

5. Figure 1A: the representation of the active and inactive RNAs in the context of the Csm complex is confusing. Is the 5' tag a separate RNA or is it that binding of the extended region leads to inactivity as the NTR has been depicted. Also, it is unclear why the Csm3 RNase activity has been highlighted in this figure if none of the reporters read out this activity.

Response: We decided to remove the mention of inactive target RNA completely from Figure 1A, as it is never used in the context of MORIARTY, although it is an essential mechanism of Csm-mediated immune response.

It is in fact important to highlight the Csm3 RNase activity here because we optimized and employed the RNase-deficient Csm3 (D30A) in MORIARTY. The RNase activity of Csm3 destroys the target RNA, which abrogates the cOA synthesis and collateral DNase activity of Csm1 and thus the virus-induced fluorescence signal.

6. End of results paragraph 2, sentence “These results show that detection of viral RNA by MORIARTY can be carried out in multiple buffer conditions.” This sentence overstates the results: the target RNA used for these experiments has not been described as virally derived and only one concentration of target RNA has been tested. Also, it appears that multiple buffer conditions are possible, but only if Mn is present.

Response: We revised the sentence to “These results show that MORIARTY can produce target-induced signals under multiple buffer conditions.”

As we showed in Figures 1 and S1, fluorescence signals can be detected in a Mn alone, Mg alone or a Mn plus Mg condition, as a result of either one or two active sites.

7. Switch Figure 1B top right with Figure S1 middle row middle panel so that ATP concentrations are consistent throughout top row of Figure 1B.

Response: The figures are now completely revised with this comment in mind.

8. Figure 1C, top left panel: please clarify in the figure itself if this is the condition without the T7 promoter resulting in no signal is observed for either reporter.

Response: This panel is now clearly labeled.

9. Figure captions should not include results interpretation; see Figure 1C caption as an example.

Response: We revised all figure captions to reflect this change.

10. Figure S2: comparison between conditions in a single row would be simpler if data were plotted on the same graph with each ATP concentration as a different color.

Response: We attempted this idea but found that six curves in a single plot with two different Y-axes make it too crowded. We hope the revised panel style still makes the trends clear.

11. SARS-CoV-2 RNA design: it would be helpful to discuss the level of sequence conservation at this site and whether given the sequence constraints of this system whether there are other sites that could be selected in case mutations accumulate in this location.

Response: It is a good idea for us to address the choice of targets. As LICsm is completely inactive when target contains 4-8 nucleotides complementary to the 5'-tag sequence in crRNA, we searched such sequences in SARS-CoV-2 genome and found no site with full 8-nt but 96 sites with 4-nt, representing <1% of the genome. This indicates very little restriction in target site selection. We added this point to the Discussion.

12. Discussion section: it is incorrect to reference the sensitivity of SHERLOCK as 50 fM. The sensitivity of Cas13 detection was observed to be this, but the SHERLOCK method includes both amplification and Cas13-based detection. I would also caution comparing MORIARTY's sensitivity to these earlier publications as target RNAs were different and there are more recent publications that could serve as a more relevant comparator.

Response: We revised the statement to reflect the fact that LwCas13a has sensitivity around 50 fM in **an amplification-free setting**.

13. The authors should test or add discussion as to whether or not their amplification-free method would detect any of the patient samples in order to convey in what contexts this amplification-free method could be used.

Response: We now able to detect patient samples directly with three targets. When S0, S7, S8 were simultaneously used to target S mRNA in a multiplexing MORIARTY reaction, we reached sensitivity of 0.5 fM in vitro transcribed S mRNA or ~2000 copies / ul SARS-CoV-2 RNA. Finally, we were able to direct detect patient samples when Ct values were low.

Reviewers' Comments:

Reviewer #1:

Remarks to the Author:

-

Reviewer #2:

Remarks to the Author:

I thank the authors for taking the time to revise and update their manuscript. I applaud the authors on their thoughtful revised introduction and discussion, repeat of essential experiments, and revamped figures. I now feel that this paper is ready for publication pending a few, very minor suggestions to improve clarity for a broader scientific audience. I have listed these suggestions below:

1. I appreciate the authors correcting their statements about antigen and serology tests. However, I believe the revised sentence (lines 40-41) that antigen tests require prior production of antibodies could be misinterpreted. I would recommend clarifying this by indicating that these antibodies are to be incorporated into the test, so that a general reader does not believe the antibodies are coming from the patient sample, but that the antibodies are within the test to detect viral antigens in the patient sample.
2. In line 46, when comparing PCR to antigen and serology tests, I would not go as far as say PCR tests are rapid (especially when compared to antigen and serology tests). Also, I agree with the authors that PCR tests can be made available prior, but only if the pathogen/pathogen sequence is known.
3. The authors state in the caption of Figure 1B and C that data shown is background-corrected. I would label the y-axes as such (i.e. Background-corrected fluorescence) to increase clarity and avoid confusion with later figures that include the water control and are not background corrected, but currently have the same axis title.

Responses to REVIEWER COMMENTS

Reviewer #2:

I thank the authors for taking the time to revise and update their manuscript. I applaud the authors on their thoughtful revised introduction and discussion, repeat of essential experiments, and revamped figures. I now feel that this paper is ready for publication pending a few, very minor suggestions to improve clarity for a broader scientific audience. I have listed these suggestions below:

1. I appreciate the authors correcting their statements about antigen and serology tests. However, I believe the revised sentence (lines 40-41) that antigen tests require prior production of antibodies could be misinterpreted. I would recommend clarifying this by indicating that these antibodies are to be incorporated into the test, so that a general reader does not believe the antibodies are coming from the patient sample, but that the antibodies are within the test to detect viral antigens in the patient sample.

Response: to avoid misinterpretation, we revised the statement to “Antigen-based tests, although effective and rapid, requires prior manufacture of antibodies to be incorporated into the test and usually lack high sensitivity.”

2. In line 46, when comparing PCR to antigen and serology tests, I would not go as far say PCR tests are rapid (especially when compared to antigen and serology tests). Also, I agree with the authors that PCR tests can be made available prior, but only if the pathogen/pathogen sequence is known.

Response: agreed. We removed the word “rapid”.

3. The authors state in the caption of Figure 1B and C that data shown is background-corrected. I would label the y-axes as such (i.e. Background-corrected fluorescence) to increase clarity and avoid confusion with later figures that include the water control and are not background corrected, but currently have the same axis title.

Response: We revised the labels in Figure 1.